# Towards Realistic Example-based Modeling via 3D Gaussian Stitching

## Abstract

Using parts of existing models to rebuild new models, commonly termed as example-based modeling, is a classical methodology in the realm of computer graphics. Previous works mostly focus on shape composition, making them very hard to use for realistic composition of 3D objects captured from real-world scenes. This leads to combining multiple NeRFs into a single 3D scene to achieve seamless appearance blending. However, the current SeamlessNeRF method struggles to achieve interactive editing and harmonious stitching for real-world scenes due to its gradient-based strategy and grid-based representation. To this end, we present an example-based modeling method that combines multiple Gaussian fields in a point-based representation using sample-guided synthesis. Specifically, as for composition, we create a GUI to segment and transform multiple fields in real time, easily obtaining a semantically meaningful composition of models represented by 3D Gaussian Splatting (3DGS). For texture blending, due to the discrete and irregular nature of 3DGS, straightforwardly applying gradient propagation as SeamlssNeRF is not supported. Thus, a novel sampling-based cloning method is proposed to harmonize the blending while preserving the original rich texture and content. Our workflow consists of three steps: 1) real-time segmentation and transformation of a Gaussian model using a well-tailored GUI, 2) KNN analysis to identify boundary points in the intersecting area between the source and target models, and 3) two-phase optimization of the target model using sampling-based cloning and gradient constraints. Extensive experimental results validate that our approach significantly outperforms previous works in terms of realistic synthesis, demonstrating its practicality.

## 1 Introduction

As we all know, 3D scenes typically contain multiple 3D objects composed of various parts. Example-based modeling Funkhouser et al. (2004) is a technique that involves combining different parts from different objects to create new ones. This is a common tool in Computer Graphics (CG) modeling, where objects are designed in a non-realistic CG fashion. In this paper, we consider realistic example-based modeling, where all parts are captured from the real world, as shown in Fig. 2. This task becomes prominent with the emergence of Neural Radiance Fields, which enables photorealistic 3D reconstruction and rendering.

Among the various approaches designed for 3D modeling from multiple neural fields, a portion of the research Gao et al. (2023); Liu et al. (2023b) is devoted to the inverse rendering process to achieve consistent lighting and shadowing. But these methods rarely consider a situation where the harmonious and seamless effect is required for merging or unifying two or more neural fields. SeamlessNeRF Gong et al. (2023) is the first work to tackle seamless merging, attempting to address the consistency problem by propagating gradients on synthesis cases. Nonetheless, due to its implicit grid-based representation, SeamlessNeRF can neither achieve fine-grained editing (e.g. the face in the *Santa* case in Fig. 1) under real-world cases nor provide an interactive workflow in real-time. Additionally, its gradient-based strategy can produce significant artifacts (see Fig. 9) and fails to propagate structural characteristics when the condition becomes more complex (e.g., the *bottle* in the left-upper corner in Fig. 2). Therefore, achieving a harmonious and photorealistic stitching result on real-world data remains an unsolved challenge that needs further exploration.

To address the limitations mentioned above, we propose a new method for interactive editing and stitching multiple parts using explicit shape representation in 3D Gaussian Splatting. Our method has two significant advantages. First, its point-based representation enables fine-grained editing, allowing for detailed appearance optimization and the removal of artifacts. Second, its rasterizer pipeline provides a real-time interactive editing environment. Due to the discrete and irregular nature of 3D-GS, it is not feasible to conduct gradient propagation as SeamlessNeRF. Thus, we introduce a novel sampling-based optimization strategy that can seamlessly propagate not only color tones but also structural characteristics. Our evaluation benchmarks are primarily derived from real-world scenes, demonstrating our superior ability to handle complex cases.

More specifically, our pipeline takes multiple scenes as input, containing source and target objects represented by 3DGS. We then carefully segment these objects and apply rigid transformations in order to create a semantically meaningful composite in 3D space. An intersection boundary region between the objects is also identified before blending. The next is the key step in our process which aims to optimize the appearance of the target objects so that their texture and color match those of the source object. We achieve this by using a two-phase optimization scheme: the first phase involves sampling-based cloning (S-phase), and the second phase involves clustering-based tuning (T-phase). During the S-phase, the target field is optimized using a heuristic sampling strategy that considers the structural characteristics at the boundary. Additionally, an efficient 2D gradient constraint is applied to preserve the original texture content of the target field. However, optimizing solely with S-phase may lead to the appearance of artifacts or unintended color features that do not fit with the overall composite. Therefore, we address this issue with T-phase, where we utilize a pre-calculated feature palette derived from the source field through aggregation and clustering. Subsequently, this palette is applied to tune the target field. It is important to note that the two-phase optimization is a joint procedure, where losses from the S-phase are always maintained while losses from the T-phase are added later during optimization.

In summary, our method makes the following contributions:

- The first work to use 3D-GS for realistic and seamless part compositing, enabling real-world example-based modeling.
- A novel sampling-based optimization strategy is proposed, with which not only the texture color but also the structural characteristics can be propagated seamlessly.
- A user-friendly GUI is carefully designed to support an interactive workflow of the modeling process in real time.

## 2 RELATED WORK

### 2.1 EXAMPLE-BASED SEAMLESS EDITING

Seamless editing, particularly in the context of example-based image and texture synthesis, is a well-studied editing technique in computer graphics and image processing. As for textures, example-based texture synthesis Wei et al. (2009); Efros & Leung (1999) intends to seamlessly create textures at any size from exemplars, which has been widely employed in contemporary graphics pipelines and game engines. In 2D image synthesis, patch-based synthesis techniques have been widely researched to seamlessly combine visually inconsistent images Pérez et al. (2023); Darabi et al. (2012). Meanwhile, Kwatra et al. (2005) introduced "Texture Optimization," which transfers photographic textures to a target image for example-based synthesis. To facilitate structural image editing tasks, "PatchMatch" Barnes et al. (2009) found approximate nearest-neighbor correspondences between patches in images for seamless image region reshuffling. In terms of seamless editing in 3D objects, Rocchini et al. (1999) and Dessein et al. (2014) propose methods for stitching and blending textures on 3D objects, respectively, while Yu et al. (2004) use the Poisson equation to implicitly modify the original mesh geometry via gradient field manipulation. Additionally, example-based modeling can also generate novel models from parts of existing models Funkhouser et al. (2004), allowing untrained users to create interesting and detailed 3D designs, such as city building Merrell (2007), things arrangements Fisher et al. (2012), mesh segmentation Katz et al. (2005), and merging Kreavoy et al. (2007). Recently, deep learning methods have leveraged generative models to generate diverse instances from a single exemplar Wu & Zheng (2022); Li et al. (2023b) or a cluster of examples Zhang et al. (2023). Definitely, the example-based methodology is a valuable tool for creating diverse and novel content, which can reduce the workload for the artists or can be leveraged

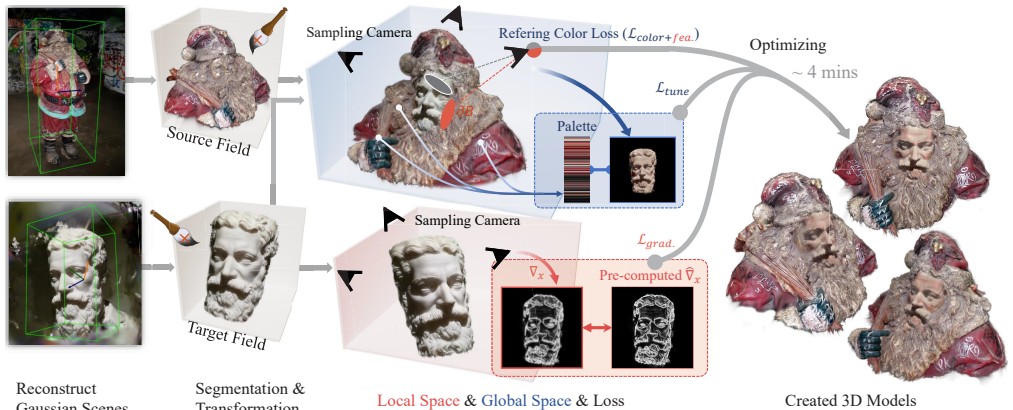

Figure 1: Overview of our framework. Our novel pipeline provides an interactive editing experience and has real-time previewing capabilities to visualize the optimizing process, allowing for the seamless and interactive combination of multiple Gaussian fields.

by procedural content generation programs. In our work, we combine this valuable idea with the advanced technique of 3DGS to create content directly from the real world.

## 2.2 NEURAL SCENE COMPOSITION

Neural scene composition primarily involves the synthesis of multiple neural objects represented by neural fields, such as free-viewport video Lin et al. (2022); Zhang et al. (2021); Wang et al. (2023), autonomous driving Ost et al. (2021); Kundu et al. (2022); Tancik et al. (2022); Fu et al. (2022); Zhou et al. (2023); Yang et al. (2023) and scene understanding Kerr et al. (2023); Shuai et al. (2022); Yang et al. (2021); Wu et al. (2022). And for those composition tasks with multiple pre-trained models, mesh scaffold Yang et al. (2022); Yariv et al. (2023) or texture extraction Tang et al. (2023b); Chen et al. (2023b) from the neural field are preferred to achieve higher render speed or rather fine-grained control. This type of work acts as a "bridge" between neural and traditional representations in order to improve performance using the classical graphics pipeline. A small portion of the work focuses on creating a mixed render pipeline for neural 3D scene composition tasks, combining traditional render techniques like ray tracing Qiao et al. (2023), shadow mapping Gao et al. (2024), and ambient occlusion Gao et al. (2023). There are also a few works that focus on creating a compositional scene with generative models like diffusion models Po & Wetzstein (2023).

None of those works except Neural Imposter Liu et al. (2023a) and SeamlessNeRF Gong et al. (2023) focus on example-based modeling by stitching multiple part NeRFs. However, part objects in Neural Imposter are just placed together without any appearance blending, which cannot support a general case of 3D modeling. SeamlessNeRF achieved harmonious results on a small-scale synthesis dataset, making it the first work to discuss seamless example-based modeling with neural techniques today. However, SeamlessNeRF cannot handle real-world cases when the condition becomes more complex, nor can it perform interactive editing, which is commonly required in example-based modeling. On the contrary, our approach overcomes these limitations, performs well in real-world scenarios, and supports interactive editing using Gaussian fields.

## 2.3 3D GAUSSIANS

3D Gaussian Splatting Kerbl et al. (2023) is a point-based rendering method that has recently gained popularity Yang et al. (2024b); Huang et al. (2024); Liang et al. (2023); Tang et al. (2023a); Chen et al. (2023a); Yang et al. (2024a); Li et al. (2024) due to its realistic rendering and significantly faster training time than NeRFs. Compared to the implicit representation of NeRF, 3DGS is more advantageous for editing tasks. The superior advance lies in the fact that, unlike previous work that embedded an object in a certain neural field (e.g., learnable grid or MLP network), once clusters of Gaussians are optimized, they can be easily fused together and fed into the rasterizer. The 3DGS pipeline was born with an intrinsic property suitable for composition.

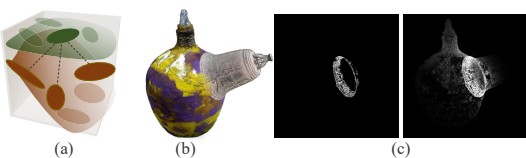

Figure 2: Our method can seamlessly stitch multiple 3D Gaussian fields together interactively, resulting in new, highly detailed, and realistic objects. All of the geometric parts or models are derived from the BlendedMVS Yao et al. (2020) and Mip360 Barron et al. (2022) datasets.

## 3 SEAMLESS GAUSSIANS

Our approach starts with segmenting interesting parts from pre-trained Gaussian scenes. After acquiring target and source models represented by Gaussians, we carefully transform them to obtain a semantically meaningful composite. Then we optimize the target objects to achieve a harmonious composite through a two-phase (sampling-based cloning and clustering-based tuning) scheme. All these processes can be run interactively and previewed in real-time with our well-tailored GUI.

### 3.1 SEGMENTING AND TRANSFORMING GAUSSIANS

Segmentation is the first step in example-based modeling, which involves picking out interesting parts as the components of the final artwork. Previous works have performed this task by providing guidance using 2D mask Cen et al. (2023); Mirzaei et al. (2023) or injecting semantic label Kerr et al. (2023) into a neural field. Now, benefiting from Gaussian representation (resembling point cloud), segmentation can become more practical at a finer-grained level. In our pipeline, we show that a combination of a simple bounding box and a user brush can work very well for a clean mask (see Fig. 15). For instance, we can mask the *sculpture* with a brush to match the shape of *Santa*'s face (see Fig. 1).

Figure 3: For a Gaussian point in the target field, its (a) K-nearest neighbors in the source field can be leveraged to justify whether this point belongs to the intersection boundary region. We use the boundary of (b) as an example to demonstrate the effectiveness of this strategy, as shown in (c).

(a)    (b)    (c)

Transformation aims at placing multiple interesting parts $\mathcal{G}_i$ represented by Gaussians to form a semantically meaningful composite $\mathcal{M}$, which can be denoted as:

$$\mathcal{G}_i^{\text{global}} = F(\mathcal{G}_i^{\text{local}} | \hat{\mathbf{q}}_i, \mathbf{t}_i, s_i), \quad \mathcal{G}_i \in \mathcal{M} \quad (1)$$

where $F$ is the rigid transformation applied on one part of Gaussians with rotation $\hat{\mathbf{q}}_i$ (represented in quaternion), translation $\mathbf{t}_i$, and scale $s_i$, transforming the part from its local space to the global space. Specifically, the partial attributes of each $\mathcal{G}$ should be modified, which includes position $\mathbf{x}$, scaling $\mathbf{s}$, rotation $\mathbf{q}$ (in quaternion), and feature $\mathbf{f}$ (represented as spherical harmonics). The position and scaling can be performed trivially, while the transformed rotation $\mathbf{q}'$ and feature $\mathbf{f}'$ can be expressed as:

$$\mathbf{q}' = \mathbf{q}\hat{\mathbf{q}}, \quad \mathbf{f}' = M_{bands}(\mathbf{f} | \hat{\mathbf{q}}), \quad (2)$$

where $M_{bands}$ means we use a set of matrices to rotate each band of SH coefficients introduced by Ivanic & Ruedenberg (1996).

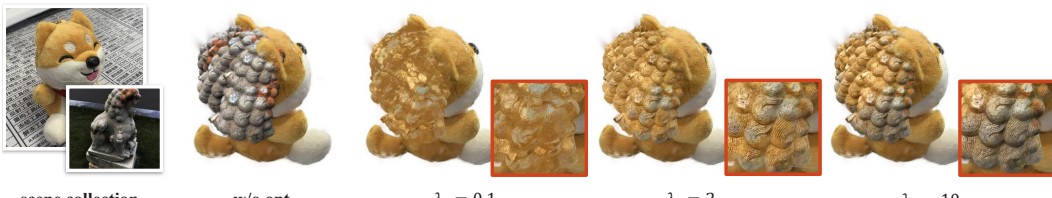

Figure 4: Ablation study on the color loss in the S-phase. Without color loss, the propagation is inefficient and will not begin. The cases shown above have been running for more than twice as long, but they are still trapped in insufficient propagation. It is because, without color loss, only a small number of points' features need to be updated at first, as opposed to shared weights in an MLP applied to all points. That minor "forces" cannot drive the overall minimization of the gradient loss.

## 3.2 BOUNDARY CONDITION BY KNN ANALYZING

After transformation, certain points in one field approach another field (see Fig. 3), forming intersection boundary regions between all Gaussians. For the sake of simplicity, we will use two Gaussians, source field and target field, to demonstrate our approach.

Before optimization, the boundary points in the target field must be identified, as this is the critical and initial condition for harmonization. For each Gaussian point in target field $\mathcal{T}$, we search its K-nearest neighbors in source field $\mathcal{S}$, which can be denoted by:

$$\{b_i\}_K = K\underset{\mathcal{S}}{N}N(a), \quad a \in \mathcal{T}, b_i \in \mathcal{S} \tag{3}$$

where $a$ is a point in the target field, and $b_i$ is a point in the source field. Whether a point $a$ belongs to boundary $\partial B$ can be identified as $a \in \partial B$ iff.:

$$\frac{1}{K}\sum_i^K |b_i - a| < \beta \quad \text{and} \quad o(a) > \tau, \tag{4}$$

where $o(a)$ is the opacity of that Gaussian point, $|b_i - a|$ is the Euclidean distance between $b_i$ and $a$. $\tau$ and $\beta$ are thresholds and we empirically set $\tau$ to 0.95, $\beta$ to $0.05 \times L$. $L$ is the size of the composite. (e.g. measured by the bounding box). An additional method for a better boundary condition on real-world data is that we discard outliers in both fields (e.g. some Gaussian points are far from the others, which may occur in some scenes).

We calculate referenced features for these boundary points in order to confirm the boundary condition. For each $a \in \partial B$, its target feature is:

$$\hat{\mathbf{f}}(a) = \frac{1}{K}\sum_i^K \mathbf{f}'(b_i), \quad a \in \partial B, b_i \in K\underset{\mathcal{S}}{N}N(a) \tag{5}$$

where $\mathbf{f}'(b_i)$ is the feature of $b_i$ after transformation. To achieve this boundary condition, we optimize boundary points toward their target features:

$$\mathcal{L}_{feature} = \sum_{a \in \partial B} \left\| \mathbf{f}'(a) - \hat{\mathbf{f}}(a) \right\|_2^2, \tag{6}$$

where $\mathbf{f}'(a)$ is the feature of $a$ and we directly apply this loss on SH coefficients.

| scene collection | w/o opt. | $\lambda_1 = 0.1$ | $\lambda_1 = 2$ | $\lambda_1 = 10$ |

Figure 5: ablation study on the effectiveness of gradient loss for different weights. Experiments show that higher weights can help to preserve more content while preventing harmonization.

## 3.3 SAMPLING-BASED CLONING

We propose sampling-based cloning as our "S-phase" in optimization. The core idea is how to seamlessly propagate the style in boundary through the remaining points in the target field while preserving its rich content. In contrast to a regular grid suitable with a gradient-based strategy in SeamlessNeRF Gong et al. (2023), Gaussian points are irregularly and discretely distributed in 3D space. As a result, alternative approaches need to be explored. A straightforward idea is that given a point in target field $\mathcal{T}$, one can calculate the feature difference between that point and its neighbors in $\mathcal{T}$, resembling "Laplacian coordinates". Then, one can use that "difference" as the regularizer while minimizing $\mathcal{L}_{feature}$. However, this naive approach may fail even before propagation begins (see Fig. 4). Furthermore, the boundary's structural characteristics (such as the *bottle-bell* intersection in the right-upper corner of Fig. 9) necessitate seamless cloning, which significantly improves the stitching quality.

Hence, we propose an effective sampling strategy to explicitly propagate features for each remaining point outside the boundary. The core idea lies in the way of searching several "driven points" for a candidate. The color of the candidate is driven by those points. For each $a \in \mathcal{T} - \partial B$, the optimizing target of its color in direction $\mathbf{d}_a$ is:

$$\hat{\mathbf{f}}(a, \mathbf{d}_a) = \frac{1}{K} \sum_{i}^{K} \mathbf{f}'(b_i, \mathbf{d}_b), \quad a \in \mathcal{T} - \partial B, \; b_i \in \underset{\partial B}{KNN}(\phi(a)) \tag{7}$$

where $\mathbf{f}(a, \mathbf{d}_a)$ means sampling SH color in view direction $\mathbf{d}_a$ (from point $a$ to camera), the same as $\mathbf{f}(b, \mathbf{d}_b)$. The camera centers are uniformly sampled from the surface of a sphere centered on the composite object's origin. It is important to note that the sampling strategy $KNN(\phi(a))$ maps the locations of nearby candidate points $a$-s to the correlated neighboring "driven points" and inherits the continuity of the textures from those "driven points". We use $\phi(x) = x + sin(\gamma \cdot \delta x)$ to add random effect by disturbing KNN searching (see Fig. 11), where $x$ is the position of $a$, $\delta x$ is the distance between $a$ and its nearest $b_i$ in boundary, and $\gamma$ is empirically set to 10. A larger $\gamma$ is suitable for higher structural frequencies. In this way, we can synthesize structurally aware stitching results. With equation 7, we add a color loss to the S-phase:

$$\mathcal{L}_{color} = \sum_{a \in \mathcal{T} - \partial B} \left\| \mathbf{f}'(a, \mathbf{d}_a) - \hat{\mathbf{f}}(a, \mathbf{d}_a) \right\|_2^2, \tag{8}$$

so that the color of those candidates can be optimized towards their target to achieve our explicit feature propagation.

To preserve the original rich content in $\mathcal{T}$, we present a more efficient gradient loss calculated in the local space of $\mathcal{T}$, leveraging the guidance in 2D screen space:

$$\mathcal{L}_{grad} = \sum_{x \in I} \left\| \nabla_x I^{\mathcal{T}}(p) - \hat{\nabla}_x I^{\mathcal{T}}(p) \right\|_2^2,$$
$$I^{\mathcal{T}}(p) = \mathcal{R}(\mathcal{G}_{\mathcal{T}}^{local}, p), \tag{9}$$

where $p$ is the randomly sampled camera in the local space of target field $\mathcal{T}$, $I$ is the rendered color image by rasterizer $\mathcal{R}$ of 3DGS. We pre-calculate $\hat{\nabla}_x I$ for each camera with the Sobel operator Sobel et al. (1968) before the optimization starts. We found that supervising gradients in screen space is more efficient than the straightforward one, as shown in Fig. 12.

## 3.4 CLUSTERING-BASED TUNING

While S-phase optimization is effective in preserving local color consistency, relying solely on it may lead to misaligned global appearance, such as uneven brightness, hues, and saturation (See Fig. 7). Therefore, we propose using a clustering extracted color palette to perform global tuning, which we refer to as the "T-phase" in optimization. This approach enhances the overall harmony of the composite by performing dynamic matches to a palette. To implement the T-phase, we first aggregate and cluster the color of the source field from various angles:

$$\{\mathbf{c}_i\}_N, \{w_i\}_N \leftarrow \mathcal{A}(\mathcal{G}_{\mathcal{S}}^{global}), \tag{10}$$

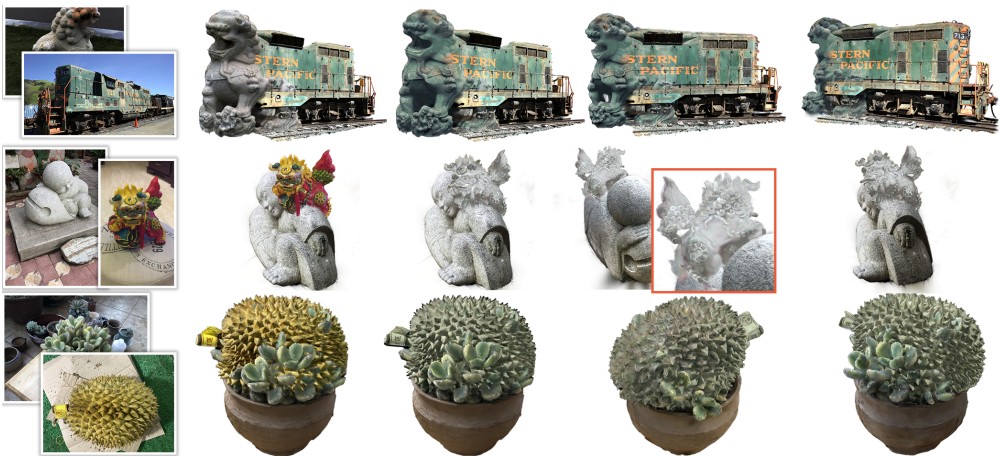

Figure 6: More real-world results from the BlendedMVS Yao et al. (2020) and Mip360 Barron et al. (2022) datasets, demonstrating that our method can produce realistic effects in real-world scenarios.

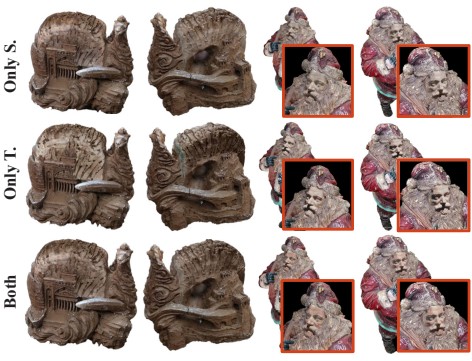

Figure 7: Ablation study on sampling-based cloning (S.) and clustering-based tuning (T.). Here, "Both" means the full scheme.

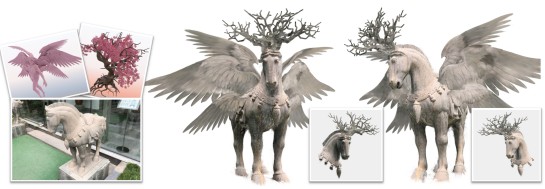

Figure 8: We demonstrate another compositing result using data from both the real world and the graphics engine. This additional case demonstrates our approach's versatility in dealing with both real and computer-generated models, validating its practical applicability. The two CG models were obtained from websites and rendered in Blender 3D on our own.

where $\mathbf{c}_i$ is the color (cluster center) in palette, $w_i$ is the sample percentage occupied by the center, and $\mathcal{A}$ stands for our aggregation algorithm.

Our approach, inspired by Li et al.'s work Li et al. (2023a), uses a streaming method to accelerate color aggregation. We start with three bins, collect color samples from a random view, and calculate the new color center for each bin by averaging the original center and new samples collected in it. The number of bins expands to accommodate far-off samples. Centers expire after 20 iterations with no sufficient votes. We repeat this process until all color centers are stable.

Once the aggregation process finishes, those color centers will form a palette (see Fig. 1). We employ the following loss in our T-phase as a pixel-wise summation:

$$\mathcal{L}_{tune} = \sum_{\mathbf{c} \in I'} w_{\chi_c} \|\mathbf{c} - \mathbf{c}_{\chi_c}\|_2^2, \ I' \leftarrow \{I_x^{\mathcal{T}}(p) | \alpha(x) > 0.95\},$$

$$\chi_c = \underset{1 \leq i \leq N}{\arg\min} \{\|\mathbf{c} - \mathbf{c}_i\|_2 - w_i\}, \tag{11}$$

where $p$ is the randomly sampled camera in the global space, and $\alpha$ is the alpha mask corresponding to $I^{\mathcal{T}}$. Both $\alpha$ and $I^{\mathcal{T}}$ are rendered by rasterizer $\mathcal{R}$. $\chi$ represents the target bin's index, and it is determined by both the distance from color centers and the probability density of bins. Our final total loss function can then be expressed as:

$$\mathcal{L}_{total} = \mathcal{L}_{feature} + \mathcal{L}_{color} + \lambda_1 \mathcal{L}_{grad} + \lambda_2 \mathcal{L}_{tune}, \tag{12}$$

| | ours | SeamlessNeRF |
|---|---|---|
| VQA average score ↑ | 0.836 | 0.662 |

Table 1: Quantitative comparison between ours and the baseline.

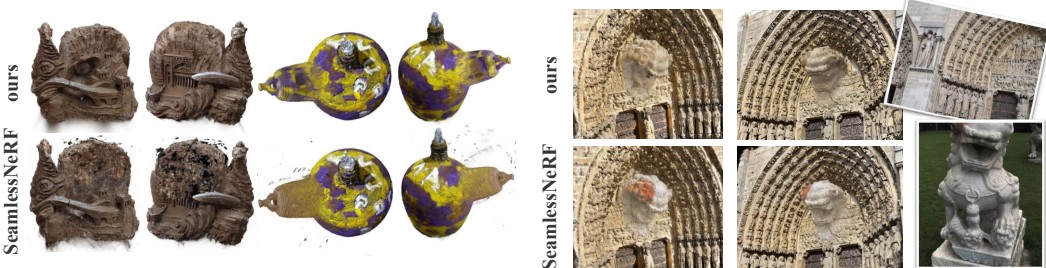

Figure 9: Comparisons between our approach and SeamlessNeRF Gong et al. (2023). Seamless-NeRF failed in all of these real-world scenarios.

where both $\lambda_1$ and $\lambda_2$ are empirically set to 2.0 in our experiments.

## 4 EXPERIMENT

To test the effectiveness and generality of our approach, we conducted experiments on a variety of fascinating 3D objects. We interactively built 21 composite results, comprising a total of 39 part models: 17 from BlendedMVS Yao et al. (2020), 4 from Mip360 Barron et al. (2022), 8 from SeamlessNeRF datasets, and 2 created by ourselves in a graphics engine. For more results or the implementation details, please refer to our supplementary materials.

### 4.1 QUALITATIVE COMPARISON

We compare our method to SeamlessNeRF Gong et al. (2023), the first and most recent work that approaches our goal. Fig. 9 depicts three comparison cases. In the first case (clay & bread), Seam-lessNeRF failed to achieve high-level geometry editability and struggled with artifacts caused by implicit representation. In the second case (bottle and bell), SeamlessNeRF failed to maintain a harmonious seamless effect due to applying the gradient-based strategy on the complex boundary. In the third case, SeamlessNeRF failed to propagate sufficient color tones due to the complex gradients in the boundary. In addition, we show that the 2D-guided style-transfer method Nguyen-Phuoc et al. (2022) cannot produce a seamless stitching effect, as shown in Fig. 10. On the contrary, ours can handle all of these situations while producing harmonious results.

### 4.2 QUANTITATIVE COMPARISON

Currently, there is neither a specialized dataset providing ground truth nor an established metric to assess the realism of a 3D model's appearance, making it challenging to evaluate the effectiveness of our approach quantitatively. Nevertheless, we force an evaluation utilizing VQA(Video Quality Assessment) methods, as outlined by Wu et al. (2023), and explored the use of 2D projection in video display for assessment purposes. Our results, presented in Tab. 1, demonstrate that our average score surpasses the baseline. For a comprehensive understanding of the quantitative experiments, please refer to our supplementary materials.

### 4.3 ABLATION STUDY

**Effectiveness of 2D Gradient Loss.** Fig. 5 depicts the effect of gradient loss at various weights. Higher weights can help to preserve more content while obstructing harmonization. Fig. 12 demonstrates that 2D gradient loss with Sobel operator is significantly more effective than the simple one mentioned in Sec. 3.3.

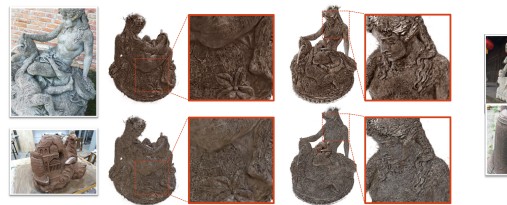
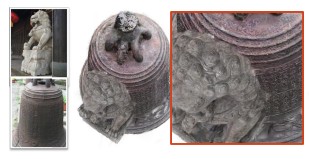
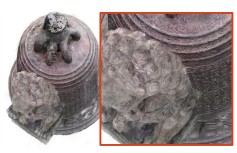

w/o $\phi$   w/ $\phi$   w/o $\phi$   w/ $\phi$

Figure 10: We show that style-transfer method fail to achieve our effect. Here, we re-implement SNeRF's strategy Nguyen-Phuoc et al. (2022) based on Gaussians to produce results above.

Figure 11: Ablation study on the impact of mapping function $\phi$ in the S-phase. Random effects make composition more realistic.

w/o sample strategy   w/ sample strategy

Figure 12: Ablation on two gradient loss. The 2D gradient supervision (upper row) is more effective than the straightforward one since it focuses on the surface instead of the whole space.

Figure 13: Ablation on keeping view-dependent effects by sampling strategy in the S-phase. With that strategy used in equation 7, the upper-view color of paint on *bell* is properly propagated.

**Functionality of S-phase and T-phase.**    We demonstrate the efficacy of our two-phase scheme in Fig. 7. The S-phase aids in seamless boundary formation, while the T-phase aids in global harmonization when only the S-phase is present.

**Effectiveness of Sampling Strategy for View-dependent Effects.**    We ablate the sampling strategy in the S-phase (see Fig. 13) to show that view-dependent effects can be properly propagated using this strategy instead of random sampling.

### 4.4 EDITOR AND APPLICATION

To enable a practical and user-friendly workflow, we created an interactive GUI editor that can control and visualize any procedure in the entire process in real-time, including Gaussian segmentation and transformation, boundary identification, and optimization (see Fig. 17 and refer to the supplementary video for more details). Our framework can generate high-fidelity and seamless results across a wide range of real-world scenarios, providing distinct advantages in the direct creation of imaginative 3D models from reality.

## 5 CONCLUSIONS AND LIMITATIONS

We have developed a highly efficient and effective interactive framework for creating realistic 3D models. The method involves stitching Gaussian components seamlessly to create a harmonious 3D model that is an accurate representation of the real world. Our approach has been tested on real-world datasets and has proved to be capable of handling complex cases with a user-friendly interface. This presents a promising avenue for example-based modeling directly from the real world.

**Limitations and Future Work.**    Currently, our work is unable to transform Gaussian models in a non-rigid manner, which may make it difficult to develop more imaginative cases. To enable a more flexible composition, we can use deformation methods such as *ARAP* Igarashi et al. (2005) in the future. Furthermore, achieving a consistent lighting effect can help improve composition quality under intense lighting.

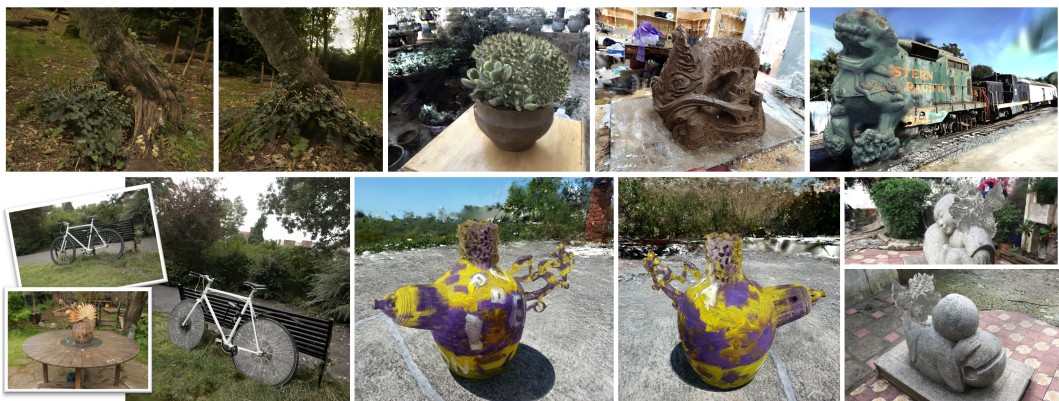

Figure 14: To demonstrate the natural appearance, we insert these composite models back into their unbounded backgrounds (the floaters are caused by the problem of 3DGS under unbounded scenes).

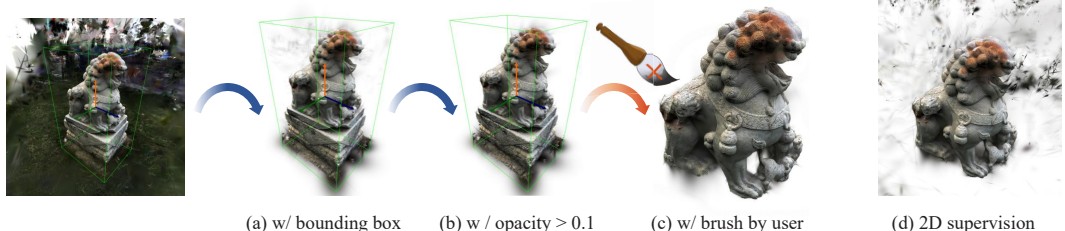

    (a) w/ bounding box     (b) w / opacity > 0.1     (c) w/ brush by user     (d) 2D supervision

Figure 15: We describe the segmentation workflow using our GUI and compare it to the result (d) from 2D mask supervision (for example, the Segment Anything Model (SAM) Kirillov et al. (2023)). To segment with SAM, we re-implement the inverse-mask Cen et al. (2023) strategy on 3DGS. A simple (a) bounding box with a (b) interactive (c) brush is demonstrated to be more practical in real-world scenes with numerous floaters. For more information, please refer to our supplementary video.

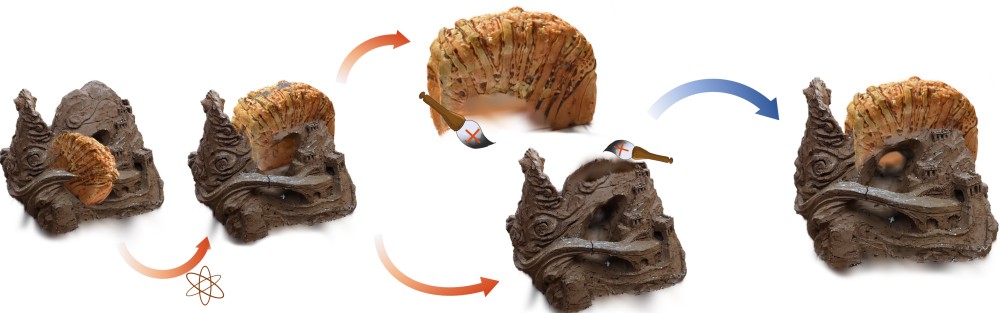

Figure 16: We describe the transformation workflow using our GUI, as well as how to remove unwanted parts during composition. Users can adjust models to create a semantically meaningful composite, and use a brush to remove unwanted parts, allowing for a more fine-grained composition. For more information, please refer to our supplementary video.

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

# APPENDIX

## A    IMPLEMENTATION DETAILS

All experiments are carried out on a single NVIDIA RTX 3090 GPU. We use the Adam optimizer for 3D Gaussian feature attributes, with learning rates of 0.02 and 0.001 for the SH feature's zero-frequency and high-frequency components, respectively. Each composition pair's optimization takes

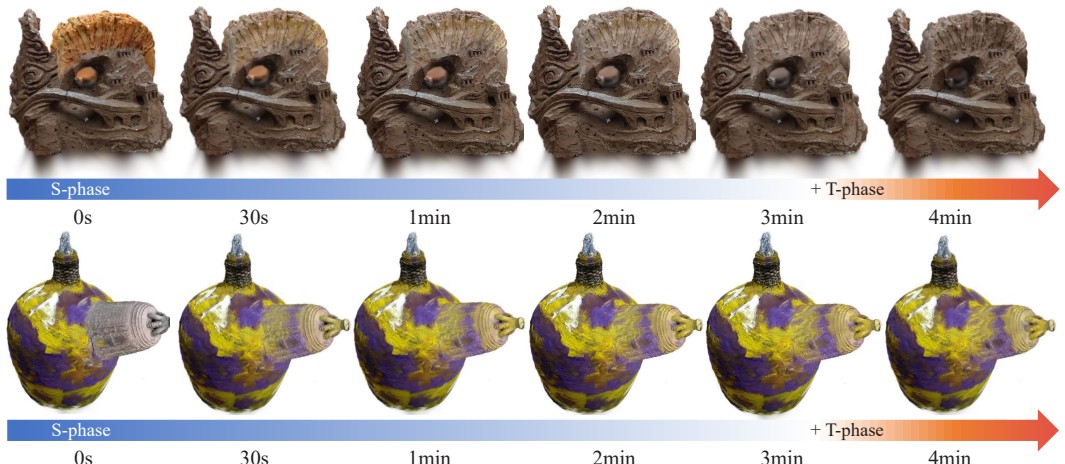

Figure 17: Visualize how our optimization gradually and efficiently converges. In our comparison, SNeRF Nguyen-Phuoc et al. (2022) takes over 10 hours, while SeamlessNeRF Gong et al. (2023) takes more than an hour. For more information, please refer to our supplementary video.

less than 5 minutes in total. During the S-phase, we sample 5,000 points as a batch from the 3D Gaussians' point cloud for each iteration rather than using the entire set; otherwise, the training speed will be slow. Throughout the KNN and palette collection, customized CUDA kernels are used to accelerate the process in less than three seconds. The entire optimization takes 6,000 iterations, consistently maintaining the loss in the S-phase and boundary conditions, with the T-phase beginning at 4,500 and continuing until completion.

## B  FAIRNESS OF COMPARISON ON REAL-WORLD DATA

We compared our method with SeamlessNeRF Gong et al. (2023), but we encountered a disparity when conducting our experiment on real-world data, prompting us to enhance the baseline performance using our approach. The discrepancy arises from the fact that SeamlessNeRF, built upon TensorRF Chen et al. (2022), was not implemented for editing scene geometry, such as segmentation and cropping. In real-world scenarios, precise masks for target objects are often unavailable, thus making the SeamlessNeRF hardly directly applied to real-world data. To compare with SeamlessNeRF on the real-world data, we utilized the interactive editing capability of our framework to generate alpha channels rendered by 3DGS Kerbl et al. (2023) to crop the target object from the background. Additionally, to ensure editing effects are based on clean density fields, we introduced a random background argumentation to mitigate artifacts during the SeamlessNeRF training process:

$$\mathcal{L}_{alphacolor} = \|w_q(c_q - \delta_q) - \hat{\alpha}_q(\hat{c}_q - \delta_q)\|_2^2 \tag{13}$$

where $w_q$ is the accumulated weights along ray $q$ in NeRF's render equation, and $\hat{\alpha}_q$ is the alpha channels generated for supervision. In the equation, $c_q$ is the color computed by our model, and $\hat{c}_q$ is the corresponding ground-truth color. The black and white background colors $\delta_q$ are randomly selected for each ray $q$ with equal possibility in our implementation. Fig. 18, shows that without this loss, too many artifacts prevent SeamlessNeRF from performing seamless editing effects. Therefore, the fairness of comparison between ours and the baseline's effects is contributed by the strength of our approach and some additional efforts, which, in turn, gives proof of our superiority.

### B.1  CHOICE OF BENCHMARK

Given the interactive nature of our method, the outcomes in all cases hinge on users' selections of compelling examples and their efforts to craft semantically meaningful results. Finding an existing dataset tailored to this specific task proved challenging. Consequently, we opted to utilize datasets such as BlendedMVS Yao et al. (2020), Mip360 Barron et al. (2022), and the synthetic data employed in SeamlessNeRF Gong et al. (2023). It is worth mentioning that while the latter dataset is

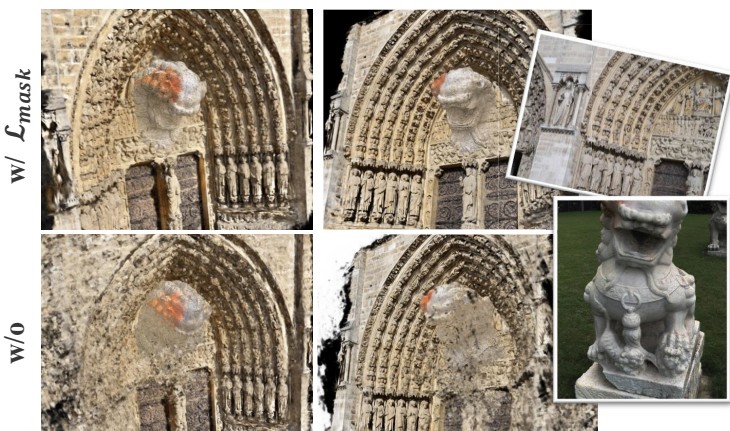

Figure 18: Improvement for SeamlessNeRF. With the help of mask loss and the mask provided by our method, artifacts are significantly suppressed, resulting in a fair comparison.

not derived from real-world sources, we have included it to underscore the discernible disparities between our approach and the baseline.

## C  MORE QUALITATIVE COMPARISON

We present a more extensive qualitative comparison, encompassing all cases in our benchmark. Direct visualization is considered to be more comprehensive than a user study. In Fig. 19, the rows (from top to bottom) represent cases numbered from 1 to 17. Cases 1-13 are derived from real-world data obtained from BlendedMVS and Mip360, while cases 14-17 originate from synthesis data used in SeamlessNeRF. The columns (from left to right) depict part models, raw composites, and two views of our method and the baseline, respectively.

## D  MORE QUANTITATIVE COMPARISON

### D.1  EVALUATING WITH VQA

The VQA (Video Quality Assessment) method acts as a tool to assess video quality, which has become increasingly essential due to the rapid increase of 2D user-generated content. Therefore, instead of evaluating the 3D models directly, we utilize VQA Wu et al. (2023) to assess the quality of the videos generated from our models. To produce coherent video sequences, we configure the camera orbit to showcase the models and ensure that the camera remains focused on the models at all times. Specifically, for results where the target field occupies a substantial space, circular camera orbits are employed to provide panoramic views, while for those occupying specific angles, spiral camera orbits are utilized (refer to our videos for visual demonstration).

**Statistic.**  Table 3 provides detailed information from the table presented in the main text. In Tab. 3, a positive number indicates that our method outperforms the baseline. The column $\Delta t$ represents the difference in the technical score, which typically relates to distortions or artifacts, while the column $\Delta a$ represents the difference in the aesthetic score, which typically reflects preferences and recommendations regarding content. It is important to note that the $\Delta a$ metric for certain cases (e.g., case 10, case 12) may not accurately reflect the true performance. This is because the VQA model struggles to comprehend seamless editing effects and instead favors situations with more diverse colors present.

### D.2  WHY NOT FID.

To compare using FID, we collected training data from the benchmark to serve as the ground truth set, enabling the identification of the distribution of realistic objects. However, the FID scores

for both methods exceeded 300, far beyond the normal range of previous generation tasks. This suggests that comparing with the FID metric makes no sense. The main reason is that the created composites themselves did not appear in any dataset. Additionally, in some cases, the backgrounds were missing, further complicating the FID algorithm's assessment.

|  | ours | SeamlessNeRF |
|---|---|---|
| average optimizing time ↓ | > 4 min | < 1 h |
| real-time adjustment | YES | NO |

Table 2: Speed Comparison between ours and the baseline.

## E  SPEED COMPARISON

Table 1 presents a concise comparison of speed, demonstrating that our method also surpasses the baseline in terms of optimization efficiency. In addition to the advantage of our method in terms of user time consumption during interactive adjustments, particularly noteworthy is the optimization speed: SeamlessNeRF requires over one hour, whereas ours takes less than 5 minutes. For visualizing the optimization process, please refer to our video.

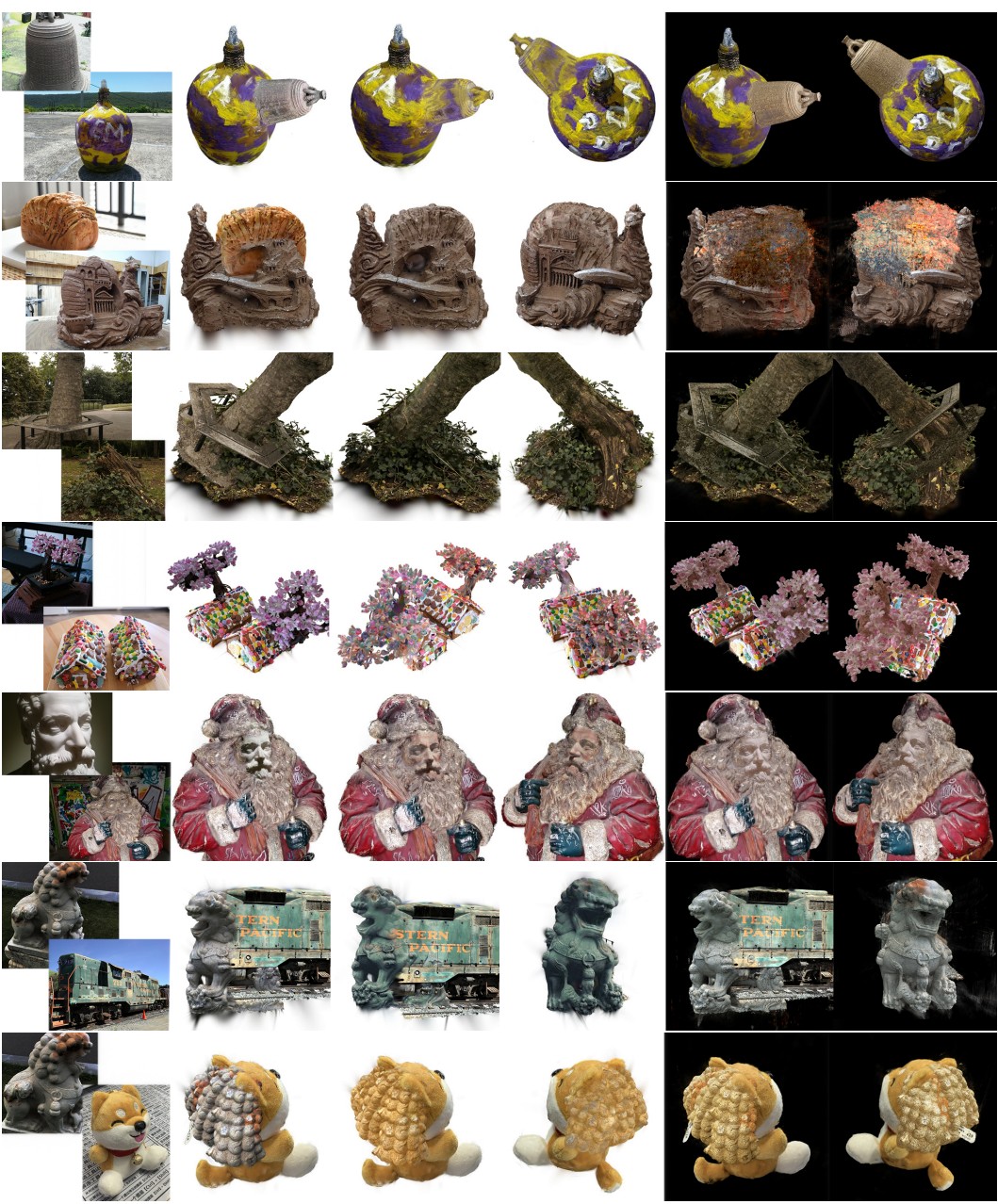

Figure 19: Case 1-7 are displayed in rows from top to bottom. The rightmost two columns present the baseline results for comparison.

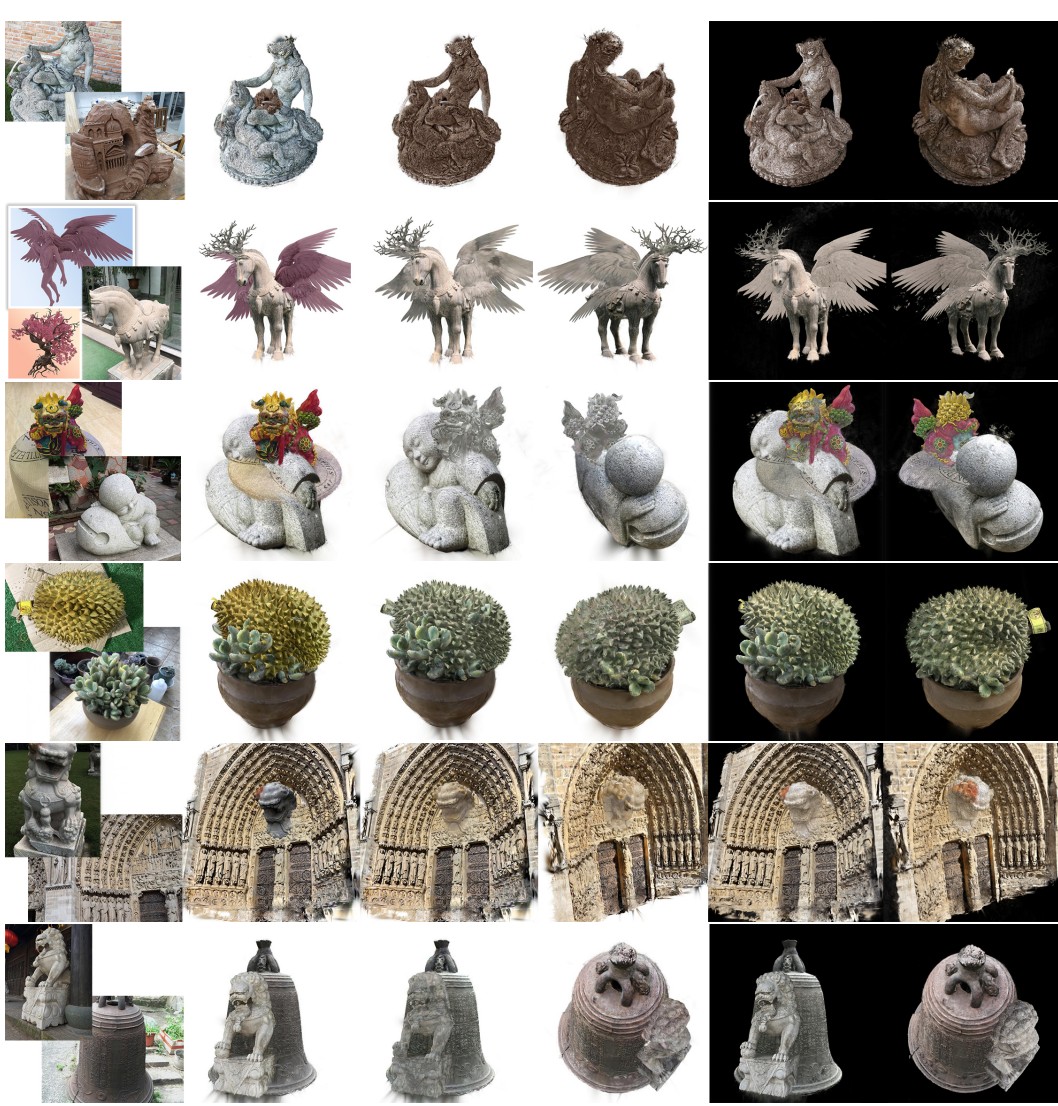

Figure 19: Case 8-13 are displayed in rows from top to bottom. The rightmost two columns present the baseline results for comparison.

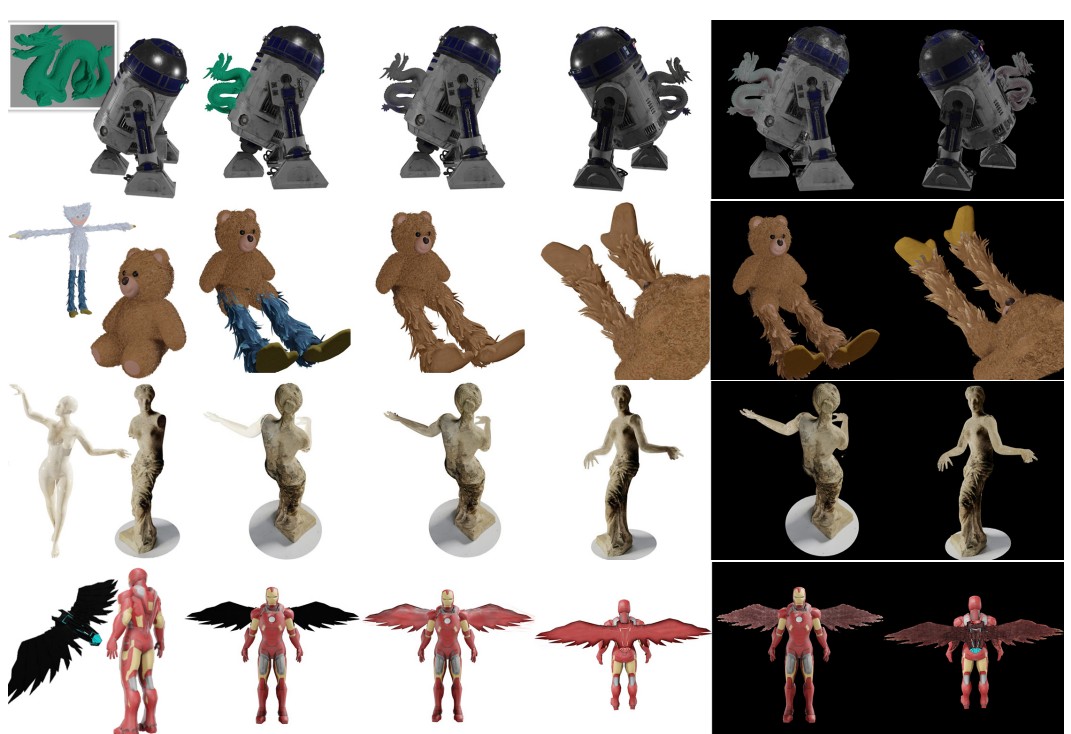

Figure 19: Case 14-17 are displayed in rows from top to bottom. The rightmost two columns present the baseline results for comparison.

| | LIVE_VQC | | KoNViD-1k | | LSVQ_Test | | LSVQ_1080P | | YouTube_UGC | |
|---|---|---|---|---|---|---|---|---|---|---|
| | $\Delta t \uparrow$ | $\Delta a \uparrow$ | $\Delta t \uparrow$ | $\Delta a \uparrow$ | $\Delta t \uparrow$ | $\Delta a \uparrow$ | $\Delta t \uparrow$ | $\Delta a \uparrow$ | $\Delta t \uparrow$ | $\Delta a \uparrow$ |
| case1 | -0.075 | +0.149 | -0.058 | +0.142 | -0.049 | +0.140 | -0.059 | +0.148 | -0.066 | +0.094 |
| case2 | +0.887 | +0.453 | +0.804 | +0.394 | +0.760 | +0.376 | +0.808 | +0.442 | +0.841 | +0.413 |
| case3 | +0.057 | +0.326 | +0.052 | +0.284 | +0.049 | +0.270 | +0.052 | +0.318 | +0.054 | +0.298 |
| case4 | +0.706 | -0.337 | +0.640 | -0.293 | +0.605 | -0.278 | +0.642 | -0.327 | +0.669 | -0.307 |
| case5 | +0.077 | +0.078 | +0.070 | +0.067 | +0.066 | +0.064 | +0.070 | +0.075 | +0.073 | +0.070 |
| case6 | +0.370 | +0.051 | +0.335 | +0.044 | +0.317 | +0.043 | +0.337 | +0.050 | +0.351 | +0.047 |
| case7 | +0.528 | +0.132 | +0.478 | +0.115 | +0.454 | +0.109 | +0.482 | +0.129 | +0.501 | +0.121 |
| case8 | +0.018 | -0.179 | +0.016 | -0.156 | +0.015 | -0.148 | +0.016 | -0.174 | +0.017 | -0.163 |
| case9 | +1.053 | +0.426 | +0.953 | +0.372 | +0.902 | +0.355 | +0.957 | +0.416 | +0.997 | +0.390 |
| case10 | +0.887 | -0.039 | +0.804 | -0.033 | +0.761 | -0.032 | +0.807 | -0.037 | +0.841 | -0.035 |
| case11 | +0.284 | +0.018 | +0.257 | -0.022 | +0.244 | +0.012 | +0.259 | -0.027 | +0.269 | -0.014 |
| case12 | +0.072 | -0.293 | +0.065 | -0.256 | +0.062 | -0.242 | +0.065 | -0.285 | +0.068 | -0.268 |
| case13 | +0.349 | -0.120 | +0.317 | -0.104 | +0.299 | -0.099 | +0.318 | -0.116 | +0.331 | -0.109 |
| case14 | +0.459 | +0.014 | +0.416 | -0.012 | +0.392 | +0.011 | +0.417 | +0.014 | +0.435 | -0.013 |
| case15 | +0.101 | -0.426 | +0.092 | -0.371 | +0.087 | -0.353 | +0.092 | -0.415 | +0.097 | -0.390 |
| case16 | +0.040 | +0.091 | +0.036 | +0.079 | +0.034 | +0.076 | +0.036 | +0.089 | +0.038 | +0.082 |
| case17 | +0.386 | +0.117 | +0.349 | +0.101 | +0.330 | +0.097 | +0.350 | +0.114 | +0.366 | +0.107 |
| average | +0.365 | +0.027 | +0.331 | +0.021 | +0.313 | +0.024 | +0.332 | +0.024 | +0.346 | +0.019 |

Table 3: Per-case Quantitative Results. We color each cell as better and worse.

