# OpenReview forum: "Towards Realistic Example-based Modeling via 3D Gaussian Stitching"
_ICLR.cc/2025/Conference — ICLR 2025 Conference Withdrawn Submission_

### Official Review · Reviewer_gZVy · 2024-10-28

**Soundness:** 3
**Presentation:** 4
**Contribution:** 3
**Rating:** 6
**Confidence:** 3

**Summary:**

This work proposes a new method for realistic 3D object composition by combining parts of existing models. Unlike prior approaches that focus on shape composition and struggle with real-world 3D objects, this method uses 3D Gaussian Splatting to achieve seamless blending of textures and structures between objects. A novel sample-guided synthesis approach allows for real-time segmentation and transformation of these objects. Additionally, a two-phase optimization process—sampling-based cloning and clustering-based tuning—ensures both local texture harmonization and global appearance consistency. Extensive experiments demonstrate that this method outperforms existing techniques like SeamlessNeRF, offering more realistic synthesis and interactive editing capabilities in real-world scenes.

**Strengths:**

* The proposed approach addresses a key gap in example-based modeling by enabling realistic and seamless stitching of 3D objects from real-world scenes, which previous techniques like SeamlessNeRF struggle with. This makes the method highly applicable for real-world applications that require detailed, cohesive compositions.
* Extensive experimentation shows that this approach achieves high-quality, harmonious blends even in challenging real-world cases, where traditional neural field blending methods fail.

**Weaknesses:**

* Although the paper’s experiments on real-world data show improvements over SeamlessNeRF, it could further benefit from a more diverse dataset, as the current scope primarily includes simple object compositions without significant lighting variation or occlusion. Expanding the dataset to more complex scenes or settings with varied lighting conditions and object types could better showcase the generalizability of the approach.
* The paper primarily demonstrates its effectiveness on objects with diffuse materials, which may limit its applicability to scenes involving highly specular or translucent objects. Complex materials with reflective or transparent properties could introduce additional challenges for the proposed method.

**Questions:**

* For larger scenes with multiple objects, how does the runtime performance scale? Are there specific optimizations that could be applied to maintain the method's real-time processing speed in more complex compositions?
* How could the proposed method be adapted for objects with highly specular or translucent materials? The experiments in the paper primarily focus on diffuse materials—would adjustments be needed to handle complex reflectance and transparency properties?

---

### Official Review · Reviewer_fSXP · 2024-10-28

**Soundness:** 3
**Presentation:** 2
**Contribution:** 2
**Rating:** 5
**Confidence:** 4

**Summary:**

This paper proposes a method for stitching 3D Gaussians, introducing KNN, T-phase, and S-phase to effectively compose and stylize 3D Gaussians. Experimental results in certain cases demonstrate that this approach outperforms baseline methods, such as seamless NeRF.

**Strengths:**

1. This paper proposes an effective method for composing pretrained 3D Gaussians, ensuring geometric accuracy and successfully addressing stylization between different objects.

2. A novel sampling-based optimization strategy is introduced to maintain the consistency of texture color between two 3D Gaussians, demonstrating the authors’ deep insight into object composition.

3. Authors develop a user-friendly GUI for composition and editing, which seems like a useful technical contribution for the proposed method.

**Weaknesses:**

1. Please use the same cases as those in seamless NeRF (e.g., Figures 4, 5, 6, and 7 from seamless NeRF) to ensure a fair comparison.

2. Providing only VQA scores and images/videos is not sufficient for a convincing evaluation. I recommend that the authors introduce a more robust evaluation metric, rather than relying solely on videos/results.

3. A user-friendly GUI is primarily a technical contribution rather than a theoretical one, even though it is emphasized in Section 1. While there is text supporting the novelty of the GUI, more emphasis is needed. Additionally, please include more detailed guidance in the supplementary material.

**Questions:**

1. Using color as a consistency supervision is sufficient to some extent, but it only achieves local statistical alignment. Have the authors considered incorporating semantic labels, such as SAM or DINO, for additional stylization supervision or evaluation?

2. Regarding the S-phase, it resembles a stylization process. Why didn't the authors follow a conventional stylization pipeline instead of using a statistical method, which may lack generalizability?

3. as for Fig.15, why implement SA3D instead of Segmeng any 3D Gaussians / Gaussian Grouping for segmentation since the latter are newer than former.

---

### Official Review · Reviewer_aLiw · 2024-11-03

**Soundness:** 3
**Presentation:** 2
**Contribution:** 2
**Rating:** 3
**Confidence:** 4

**Summary:**

The paper presents a novel method for realistic example-based modeling using 3D Gaussian Splatting (3DGS). It addresses the limitations of current methods like SeamlessNeRF by enabling interactive editing and seamless stitching of 3D models from real-world scenes. Key innovations include a real-time GUI for model segmentation and transformation, KNN analysis for boundary detection, and a two-phase optimization strategy for texture blending.

**Strengths:**

The paper is well-written, logically clear, and readable.

The proposed method is reasonable and improves the Nerf-based method by seamlessly integrating 3DGS with the real-world model.
The paper provides experimental results to support its claims. Visualization results demonstrate the effectiveness of the method.

**Weaknesses:**

1.  In Figure 13, the color propagation results are not clear. Could the authors provide examples with stronger color contrast or offer a detailed explanation of the color propagation, especially how the method handles various texture complexities?
2. The appendix briefly mentions time consumption comparisons, but could the authors elaborate on the computational demands of their method in the main text, perhaps comparing it to other existing methods in more detailed scenarios?
3. The paper mentions a GUI editor that facilitates the interactive modeling process. Could the authors provide more details or a demonstration of how the GUI editor is used, especially any features that help users manage complex scenes or models?
4. The success of the method heavily relies on the quality and complexity of the initial 3D models. Poor initial models might lead to suboptimal results, which needs further discussion in the paper.
5. The discussion of limitations is insufficient. It is recommended to add detailed pictures and discussion details of the failure of the paper's method.

**Questions:**

see the weaknesses.

---

### Official Review · Reviewer_UPfG · 2024-11-03

**Soundness:** 3
**Presentation:** 2
**Contribution:** 2
**Rating:** 3
**Confidence:** 2

**Summary:**

The paper proposes a method for interactively editing radiance fields encoded in point clouds ("Guassian Splatting"). It consists of two components: First, point clouds have to be segmented, which seems to be done mostly manually with seam-detection supported by nearest-neighbor detection across two point clouds to be stitched. It appears to me that this is not the main focus of the paper. The second component is an adaptation of Poisson image editing to radiance fields, where local differences in radiance information (encoded as spherical harmonics) is propagated along with boundary constraints. To improve realism, additional constraints are added, such as attracting the computed colors towards clusters of "colors" (actually, appearance / radiance information) in the target shape (to which a new piece is stitched), look-ups of nearby colors in the target point cloud for smoother transitions, and ad-hoc mid-frequency "noise" (function $\Phi$) for increased variability.

The resulting merged objects look quite convincing and appear more natural than competing results in the examples selected in the main paper. The appendix provides a more detailed evaluation, including quantitative results.

Limitations are not discussed (aside from the obvious limitation to merging rigid pieces, to which the paper limits itself a priori).

**Strengths:**

The idea of using differential coordinates (as in Poisson image editing, or, put alternatively, editing in higher-frequency bands of the appearance signal only) in the context of editing "Gaussian Splatting" scenes is quite appealing; if it wasn't for other reservations, I would consider this a sufficient argument for accepting the paper (I should note, however, that I am not an active researcher in this area and thus might overlook prior work; but if this is new, I would think it is a really nice and useful idea for interactive editing of such data).

The results obtained are convincing (see below for qualifications) and the interactive demo of the system shown in the supplementary material shows that this could be practically applied.

Further, the general area of making use of scene representations obtained from matching "real-world" photos by making them editable is certainly important for making the whole research area useful to artistic applications in computer graphics.

**Weaknesses:**

First of all, I have the impression that this paper is out of scope of ICLR. In terms of its approach and methodology as well as the problem it addresses, I would see this firmly within "computer graphics and interactive techniques"; it has only tangential impact on machine learning and representation learning. It might be close enough to warrant consideration, but I would see a graphics venue as a far better fit (for example, in computer graphics people would be much more interested in how this could lead to better software systems for creating better content, rather than evaluating the suitability and performance of the algorithms and data structures proposed for fitting and representing data in general).

That said, I would also seem room for improvement in positioning the paper. It seems to me that the stitching algorithm (differential appearance merging) is the main idea, while segmentation does not offer much novelty over the state of the art (see for example the long line of work of direct point cloud editing methods starting with Pauly et al.'s PointShop3D [Siggraph 2002, 2003]. More sophisticated segmentation algorithms, such as using graph-cut methods on point clouds are also known and for example already available in open source software (PCL/point cloud library). While these do not directly handle radiance data, it would be not a big step to integrate such ideas to reduce manual effort. Similarly, a long line of automatic outlier detection and removal methods could also improve this step easily (see for example "tensor voting" as a very basic approach). My point here is that the contribution of this paper is rather moderate, so it would make sense to position the paper with more emphasis on the second step (differential merging) which I personally found to be really neat.

Another issue I see (and again, probably more in writing/positioning than conceptually) is that the method mixes basic ideas (such as differential coordinates for appearance) with ad-hoc heuristics such as sinusoidal perturbations of color look-ups to create more appealing results. Differentiating between artistic and foundational ideas would make the paper stronger, in particular in the context of a rather technical venue aiming at fundamental machine learning research rather than graphics and interaction.

Finally, I am also a bit disappointed by the very short discussion of limitations in the main paper, which basically just states that the paper only considers rigid pieces; I would think that there are certainly trade-offs and mismatches in prior assumptions in the presented approach, independent of whether the geometry to be merged was also deformed prior to merging, that could be discussed more clearly. Understanding "when it breaks" is very important in order to put such ideas to use in new context.

Overall, I like the key idea, and if it is really novel, this is a strong point, but limitations in presentation and evaluation (in particular, discussion of limitations) make me skeptical; for the context of ICLR I would thus at this point be hesitant in terms of making a positive recommendation for this submission.

**Questions:**

When does the method break? Can you characterize the (or a) class of shapes where the assumptions made work well and what happens if the data does not match these assumptions?

---

### Note · Authors · 2024-11-13

I have read and agree with the venue's withdrawal policy on behalf of myself and my co-authors.